# Changes in Optical Properties of Model Cholangiocarcinoma after Plasmon-Resonant Photothermal Treatment

**Vadim D. Genin** [1,2,*], **Alla B. Bucharskaya** [1,2,3], **Georgy S. Terentyuk** [1,3], **Nikolai G. Khlebtsov** [4], **Nikita A. Navolokin** [1,3], **Valery V. Tuchin** [1,2,5] and **Elina A. Genina** [1,2]

1 Science Medical Center, Saratov State University, 83 Astrakhanskaya Str., 410012 Saratov, Russia; allaalla_72@mail.ru (A.B.B.); vetklinikanew@mail.ru (G.S.T.); nik-navolokin@yandex.ru (N.A.N.); tuchinvv@mail.ru (V.V.T.); eagenina@yandex.ru (E.A.G.)
2 Laboratory of Laser Molecular Imaging and Machine Learning, Tomsk State University, 36 Lenin's Av., 634050 Tomsk, Russia
3 Core Facility of Experimental Oncology, Saratov State Medical University, 112 Bolshaya Kazachiya Str., 410012 Saratov, Russia
4 Laboratory of Nanobiotechnology, Institute of Biochemistry and Physiology of Plants and Microorganisms, FRC "Saratov Scientific Centre of the Russian Academy of Sciences", 13 Prospekt Entuziastov, 410049 Saratov, Russia; khlebtsov@ibppm.ru
5 Laboratory of Laser Diagnostics of Technical and Living Systems, Institute of Precision Mechanics and Control, FRC "Saratov Scientific Centre of the Russian Academy of Sciences", 24 Rabochaya Str., 410028 Saratov, Russia
* Correspondence: versetty2005@yandex.ru; Tel.: +7-917-202-66-10

**Abstract:** The heating degree of the inner layers of tumor tissue is an important parameter required to optimize plasmonic photothermal therapy (PPT). This study reports the optical properties of tissue layers of transplanted cholangiocarcinoma and covering tissues in rats without treatment (control group) and after PPT using gold nanorods (experimental group). PPT was carried out for 15 min, and the temperature on the skin surface reached $54.8 \pm 1.6\ ^{\circ}$C. The following samples were cut out ex vivo and studied: skin, subcutaneous connective tissue, tumor capsule, top, center, and bottom part of the tumor. The samples' absorption and reduced scattering coefficients were calculated using the inverse adding–doubling method at 350–2250 nm wavelength. Diffuse reflectance spectra of skin surface above tumors were measured in vivo in the control and experimental groups before and immediately after PPT in the wavelength range of 350–2150 nm. Our results indicate significant differences between the optical properties of the tissues before and after PPT. The differences are attributed to edema and hemorrhage in the surface layers, tissue dehydration of the deep tumor layers, and morphological changes during the heating.

**Keywords:** gold nanorods; plasmonic photothermal therapy; optical properties of tissues; transplanted model tumor; cholangiocarcinoma

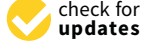



## 1. Introduction

Among the numerous challenges facing modern medicine in the 21st century, the fight against cancer occupies a special place. According to the World Health Organization, cancer is the most important cause of death and morbidity after cardiovascular diseases, with about 10 million deaths each year [1]. With many traditional treatment methods, such as surgery, chemotherapy, photodynamic therapy, radiation therapy, etc., laser plasmonic photothermal therapy (PPT) is currently tested [2–4]. The PPT relies on the accumulation of plasmonic nanoparticles in tumor tissues and their local heating under laser irradiation [5]. Accordingly, the irradiation dose can be reduced, thus reducing unwanted damage to the healthy tissues. Nanospheres, nanocubes, nanostars, nanocomposites, nanocontainers, etc., are currently used for PPT [5]. Recently, several works [6–8] have demonstrated the

high efficiency of gold nanorods (GNRs) for PPT due to their long lifetime in the bloodstream, colloidal stability, easy adjustment of plasmon resonance by changing the aspect ratio [9], and efficient conversion of light radiation into thermal energy [7]. Experimental studies [2,8] demonstrate the effectiveness of using GNRs with a plasmon resonance excitation wavelength of approximately 800 nm since it penetrates deep enough into the tissue and provides effective photothermal therapy.

Successful phototherapy of tumors sensitized with nanoparticles requires solving some problems associated with the choice of a protocol for single or multiple administration of nanoparticles, the administered dose and the distribution of tumor particles, as well as the radiation dose [10–12]. Knowledge of the optical parameters of tumor tissues is crucial for assessing the radiation dose (i.e., absorbed energy) of tumor tissues during photothermal or photodynamic therapy. This, in turn, makes it possible to correctly plan the geometry of the delivery of laser radiation into the tumor and to ensure sufficient heating of tissues located at different depths. In some cases, tumor re-growth occurs after photodynamic and photothermal suppression of its growth due to preserving even a small number of living cancer cells on the periphery of the untreated tumor [12–14].

Optical parameters of different tumor tissues have been reported previously, in particular, a detailed review is presented in [15]. In addition, changes in the optical parameters of biological tissues under heating were reported for adipose tissue [16], brain [17], skin [18], prostate [19], and liver [20]. However, those studies were carried out on whole tumor tissue, without differentiation by layers, whereas the primary accumulation of nanoparticles occurs in the blood vessels of the tumor and around them, which are localized inhomogeneously mainly in the peripheral part of the tumor. Thus, the heating of the tumor tissue is also uneven. Therefore, information about the heating of the inner layers of tumor tissue in vivo is required to optimize laser exposure.

This work aims to investigate the optical properties of the rat model tumors with GNR doping and PPT. In contrast to previous studies, we consider the temperature-induced changes in different tumor parts, including skin, sub-skin layer, capsule, tumor top, tumor center, and tumor bottom.

## 2. Materials and Methods

The GNRs (Figure 1a) were synthesized as was described earlier in [21]. The T-matrix simulations were performed for randomly oriented nanorods in water; the average particle length $L_{av}$ = 44.0 ± 8.3 nm, diameter $d_{av}$ = 10.9 ± 2.1 nm, and aspect ratio $AR_{av}$ = 4.03 ± 0.72 (Figure 1b) were derived from TEM images (transmission electron microscopy, Libra-120, Carl Zeiss, Jena, Thuringia, Germany). Simulation details are referred to in [22]. GNRs were functionalized with thiolated polyethylene glycol (molecular weight 5000, Nektar, North Hollywood, CA, USA) to prevent aggregation in blood and tissue and enhance biocompatibility [23]. The concentration of GNRs in the suspension was 400 μg/mL. Before measuring the extinction coefficient, the suspension was diluted in a ratio of 1:10. The extinction coefficient spectrum (Figure 1c) was recorded using a Specord BS-250 UV–vis spectrophotometer (Analytik Jena, Jena, Thuringia, Germany).

The animal experiments were performed at the Centre of Collective Use of Saratov State Medical University following the University's Animal Ethics Committee and relevant international agency review [24]. The rats were anesthetized with Zoletil 50 (Virbac, Val de Reuil, Normandy, France) with a dose of 0.05 mg/kg.

A model cholangiocarcinoma tumor was grown from the PC-1 cell line and transplanted by subcutaneous cell suspension injection into the scapula area of 8 outbred albino rats. The cell concentration in suspension was $2 \times 10^6$ cells per 0.5 mL of Hanks' solution.

A mature tumor includes a capsule and a stroma. The capsule surrounding the tumor is represented by connective tissue. Histological analysis of cholangiocarcinoma reveals oval-shaped tumor cells with eccentrically arranged nuclei. A significant part of the cytoplasm is occupied by large vacuoles containing mucus. Mucus accumulations are also

observed in the intracellular space. In the stroma of the tumor, its capsule and around it are newly formed blood vessels of the capillary type [25].

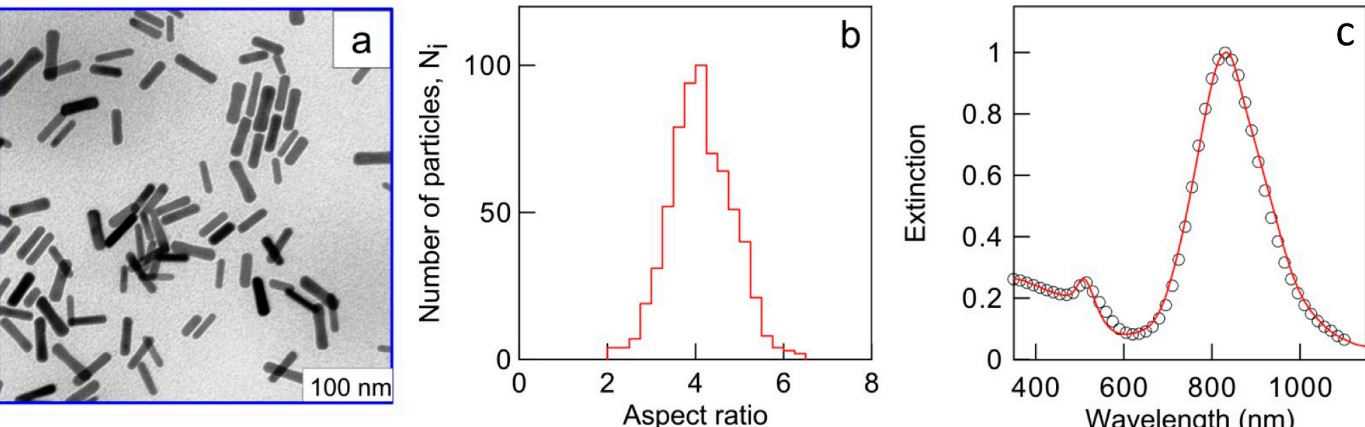

**Figure 1.** TEM image of Au nanorods (**a**), the aspect ratio histogram (**b**), and measured (black circles) and T-matrix-simulated extinction spectrum (red line) (**c**) (for details, the readers are referred to [22]).

When the tumor volume reached about 3 cm$^3$, the rats were randomly divided into two groups of four animals: a control group (without any treatments) and an experimental group that was twice intravenously injected with 1 mL of GNRs suspension 48 h and 24 h before PPT (Figure 2a).

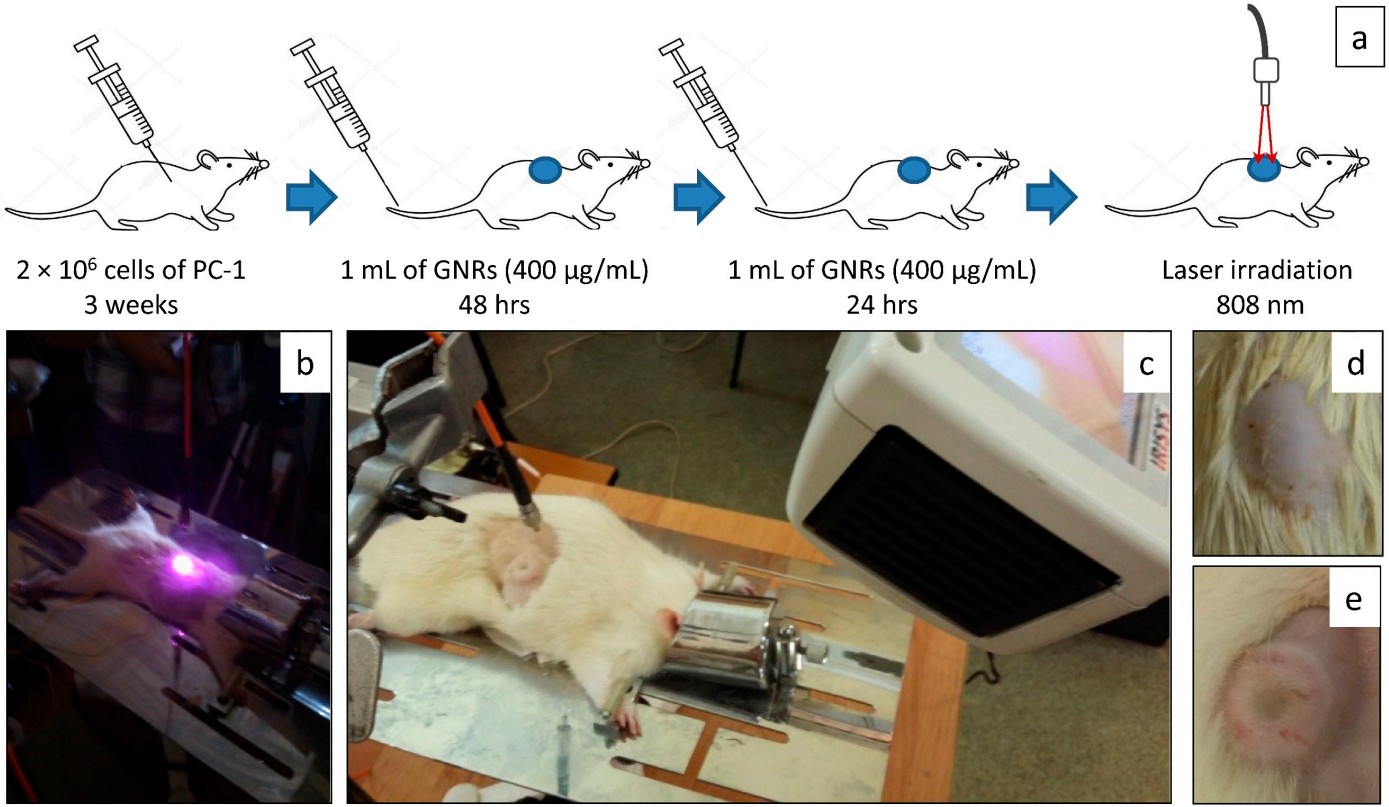

**Figure 2.** Scheme of experiment (**a**), photos of the experimental setup during PPT (**b**) and just after PPT (**c**), and photos of the tumors from the control group of the rats (**d**) and experimental group after PPT (**e**).

A diode laser LS-2-N-808-10000 (Laser Systems LLC, Saint Petersburg, Russia) with a wavelength of 808 nm and a power density of 2.3 W/cm$^2$ was used for irradiation. The duration of exposure was 15 min. The temperature of the skin surface above the tumor

was measured every 30 s using an IR visualizer IRI4010. The influence of scattered laser radiation on the results of the temperature measurements was eliminated because the infrared imager is sensitive in the spectral range from 8 to 14 μm. Images of experimental setup and tumors from the control group and from the experimental group after PPT are presented in Figure 2b–e.

The wavelength of 808 nm was chosen to match the plasmon resonance of the nanoparticles used in order to provide selective absorption of laser radiation mainly by nanoparticles (see Figure 1), which will concentrate in the tumor. Previously, we tested the accumulation of gold and the effect of PTT after twice intravenous injections of similar nanorods [8,12]. The developed method of PTT induced a pronounced regression of grafted tumors, manifested by necrobiotic changes in tumor cells and inhibition of tumor growth.

To evaluate the immediate effect of heating in vivo, the diffuse reflectance spectra were measured before and immediately after PPT using two commercially available spectrometers USB4000-Vis-NIR (Ocean Optics, Dunedin, FL, USA) in the spectral range 350–1000 nm and NIRQuest (Ocean Optics, Dunedin, FL, USA) in the spectral range 1000–2150 nm equipped with optical standard reflection/backscatter probes QR400-7-VIS/NIR (Ocean Optics, Dunedin, FL, USA) (Figure 3). The probe design included a 6-fiber leg connected to a light source (1), a single-fiber leg connected to a spectrometer (2), and a leg with 6 fibers located around 1 fiber (3), with a core diameter of 400 μm and a numerical aperture of $0.22 \pm 0.02$. The probe was fixed in a cylindrical holder (4) with a hole diameter corresponding to the diameter of the probe to provide a distance of 2 mm from the skin surface and the probe. Thus, the diffuse backscattered radiation was collected from the skin area of about 8 mm$^2$. A halogen lamp HL-2000 (Ocean Optics, Dunedin, FL, USA) was used as the light source. The spectrometer was calibrated using a reflectance standard WS-1-SL (Labsphere, North Sutton, NH, USA).

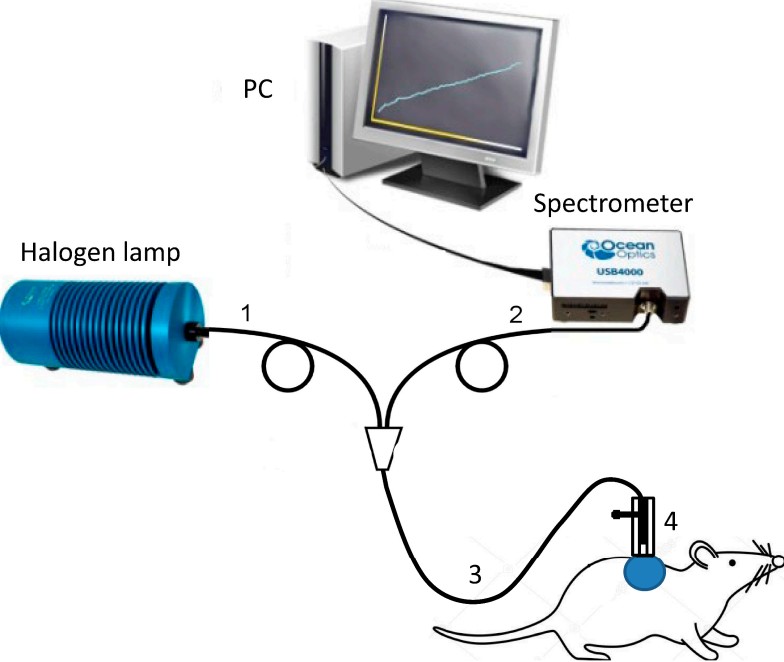

**Figure 3.** The scheme of the experimental setup for measurement of diffuse reflectance: 1 is the leg with lighting fibers, 2 is the leg with the collecting fiber, 3 is a leg with the 6-around-1 fiber bundle, and 4 is the holder of the probe.

The decapitation of the rats and tumor removal was performed immediately after PPT. The tumors were divided into several parts as follows: capsule, upper, central, and lower layers, as well as tissues above the tumor (skin and subcutaneous layer of connective tissue) (Figure 4a). The samples were placed without compression between two glass slides (Figure 4b).

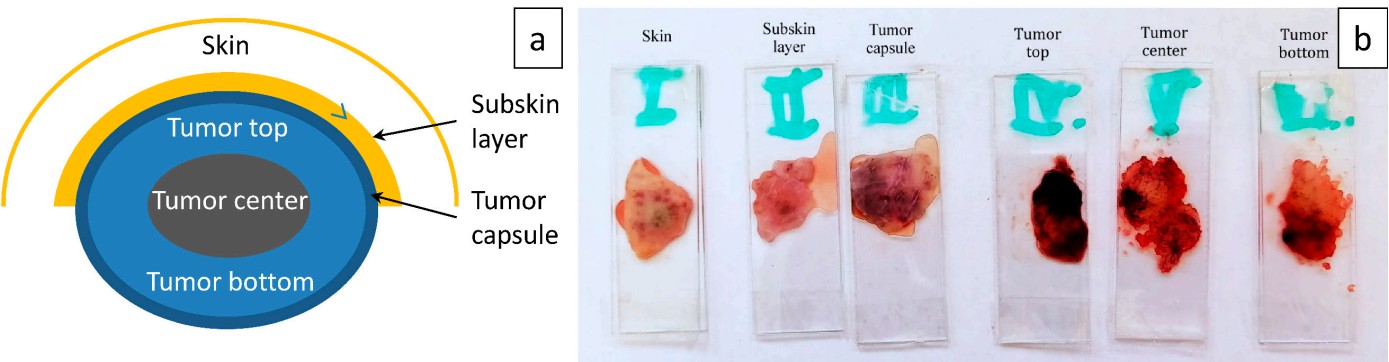

**Figure 4.** The scheme of the model tumor (**a**) and the samples of the tissue layers (**b**).

The total transmittance and diffuse reflectance spectra were measured in the wavelength range of 350–2250 nm using a commercially available spectrophotometer UV-3600 equipped with an integrating sphere LISR-3100 (Shimadzu, Kyoto, Japan). As a light source, a halogen lamp with filtering of the radiation in the studied spectral range was used. The diameter of an incident light beam on the tissue sample was $4 \times 4$ mm$^2$. Spectral measurements were carried out in the central part of the samples.

The thickness of each sample was measured with an electronic micrometer with a precision of $\pm 1$ μm. The average thicknesses and standard deviation (sd) of the samples are presented in Table 1.

**Table 1.** The thicknesses (mean $\pm$ sd, mm) of the samples from experimental and control groups.

| Groups | Skin | Subskin | Capsule | Top | Center | Bottom |
|---|---|---|---|---|---|---|
| Control | $1.46 \pm 0.39$ | $0.89 \pm 0.67$ | $0.15 \pm 0.06$ | $1.04 \pm 0.09$ | $0.52 \pm 0.32$ | $0.86 \pm 0.68$ |
| Experimental | $1.37 \pm 0.29$ | $0.52 \pm 0.23$ | $0.30 \pm 0.01$ | $0.72 \pm 0.08$ | $1.15 \pm 0.46$ | $1.37 \pm 0.25$ |

Both the absorption coefficient and the reduced scattering coefficient of the studied samples were calculated by solving the inverse problem using the inverse Monte Carlo method [26]. Averaging was carried out by groups. All data were presented in the form of mean $\pm$ sd.

## 3. Results

The averaged time dependence of the heating of the tissue surface in the experimental group is shown in Figure 5. The temperature rise reached up to $54.8 \pm 1.6$ °C. To approximate the heating kinetics, we used the empirical equation [27]:

$$T(t) = A_1 \left( 1 - \exp\left( -\frac{t}{\tau_1} \right) \right) + A_2 \left( 1 - \exp\left( -\frac{t}{\tau_2} \right) \right) + T_0, \tag{1}$$

where $A_1 = 7.5 \pm 0.7$ and $A_2 = 15.5 \pm 0.4$ are empirical constants, $\tau_1 = 0.7 \pm 0.1$ min and $\tau_2 = 7.5 \pm 0.9$ min are the constants that characterize the heating rates, and $T_0$ is the initial temperature (before heating) having the mean value of $33.8 \pm 0.3$ °C. The first and the second terms of the equation describe the kinetics of fast and slow heating, respectively.

Figure 6 shows reflectance spectra measured in vivo from the skin surface above tumors in the control group and the experimental group before and immediately after PPT. Spectra obtained from the control group did not differ significantly from the spectra obtained from the experimental group before the treatment. As it was estimated earlier in [12], the concentration of the GNPs localized in the tumor mainly in the vessels after two-time 0.4-mg intravenous injection was about $1.24 \pm 0.01$ mg/g of tissue. Thus, the nanoparticles did not affect the shape of the diffuse reflection spectra of the skin. The deviation of the spectra was caused by the features of the tissues and their blood supply in different animals.

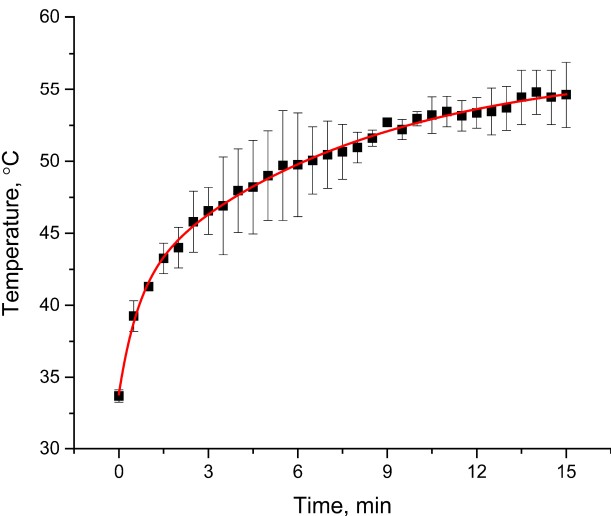

**Figure 5.** Temperature kinetics of the tissue surface in the experimental group. The symbols correspond to the averaged experimental values with sd, and the solid curve corresponds to the approximation.

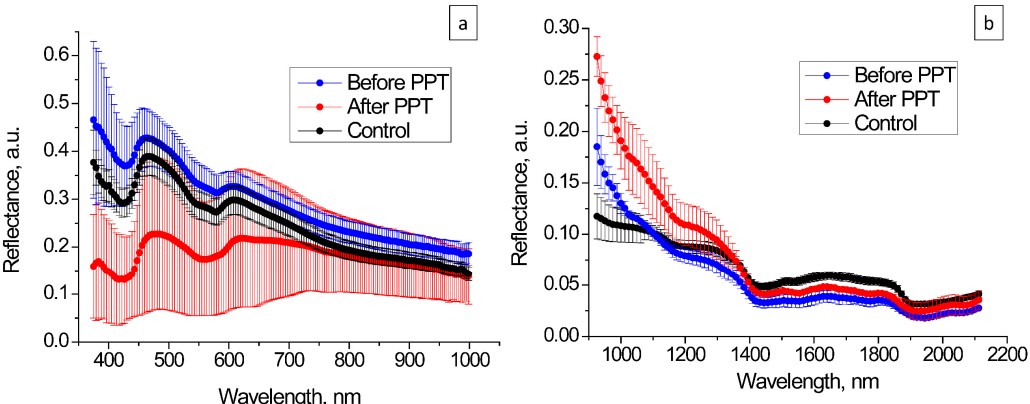

**Figure 6.** Spectra of the reflectance of the tumor before and after PPT in different spectral ranges: visible-NIR (**a**) and NIR (**b**).

In Figure 6a, a double absorption peak of oxyhemoglobin between 547 and 580 nm before the PPT can be clearly observed. It is replaced by a single peak of deoxyhemoglobin at approximately 560 nm after heating. Besides, decreasing reflectance in the short-wave range is observed due to the increase in hemoglobin absorption. It is caused by hemorrhage of the microvessels during overheating. As can be observed in in Figure 2e, the dark spot corresponds to the hemorrhage area. Characteristic bands observed in Figure 5 at the wavelengths of about 1205, 1444, and 1944 nm are caused by water absorption. They also exhibit blue shifts during the average temperature rise from 33.8 to 54.8 °C. Specifically, the water absorption band shifts from 1205 nm before the heating to 1195 nm after the heating, whereas the absorption bands located at 1444 nm and 1944 nm shift to 1437 nm and 1929 nm, respectively.

Figure 7 shows spectra of the absorption coefficient of tumor, skin, and sub-skin tissues from the control and experimental groups. These spectra demonstrate the relative content of blood and water in different layers of the objects under study. An increase in the standard deviation of the absorption coefficient observed in the area of absorption bands indicates a difference in the content of water and hemoglobin in different tissue samples.

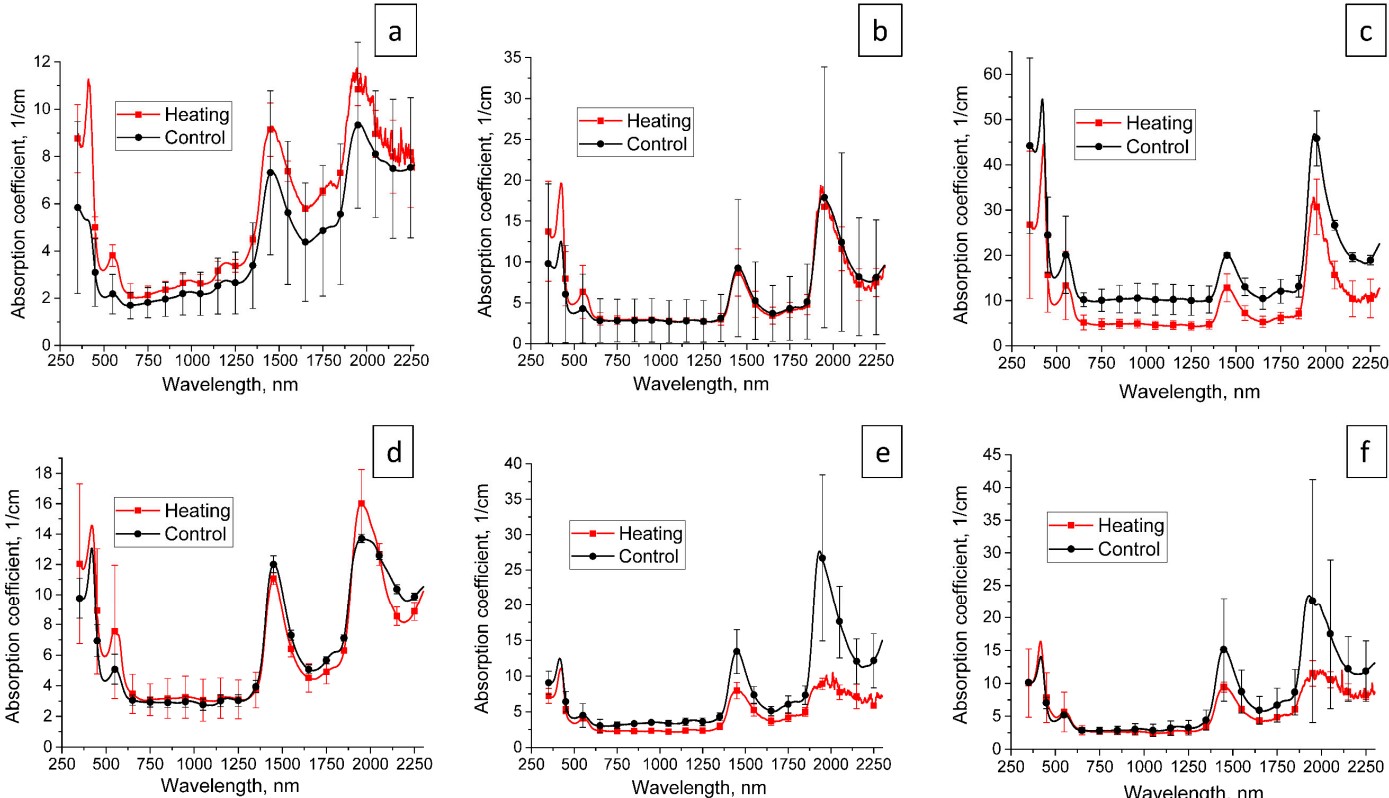

**Figure 7.** Spectra of the absorption coefficient of tumor and surrounding tissues from experimental and control groups: (**a**) skin, (**b**) sub-skin layer, (**c**) tumor capsule, (**d**) tumor top, (**e**) tumor center, and (**f**) tumor bottom.

Figure 7a,b demonstrate the increase in hemoglobin absorption bands in the skin and sub-skin tissue after heating that is in good agreement with diffuse reflectance spectra (Figure 6a). Moreover, a significant increase in the amplitude of water absorption bands is observed in the skin due to the development of edema.

A decrease in the absorption coefficient of tumor layers in the water absorption bands can be induced by tissue dehydration during PPT, which is observed mainly in both the central and bottom areas of the tumor.

Figure 8 shows the spectra of reduced scattering coefficients of the tumor and covering tissues from the control and experimental groups. The effect of the deviation of the spectral dependence of the scattering characteristics from the monotonic one is explained by an increase in the influence of the imaginary part of the complex refractive index of the scattering centers in the absorption bands of water and hemoglobin [28,29]. We can see the slight changes in the slope of spectra in the visible spectral range between the control and experimental groups.

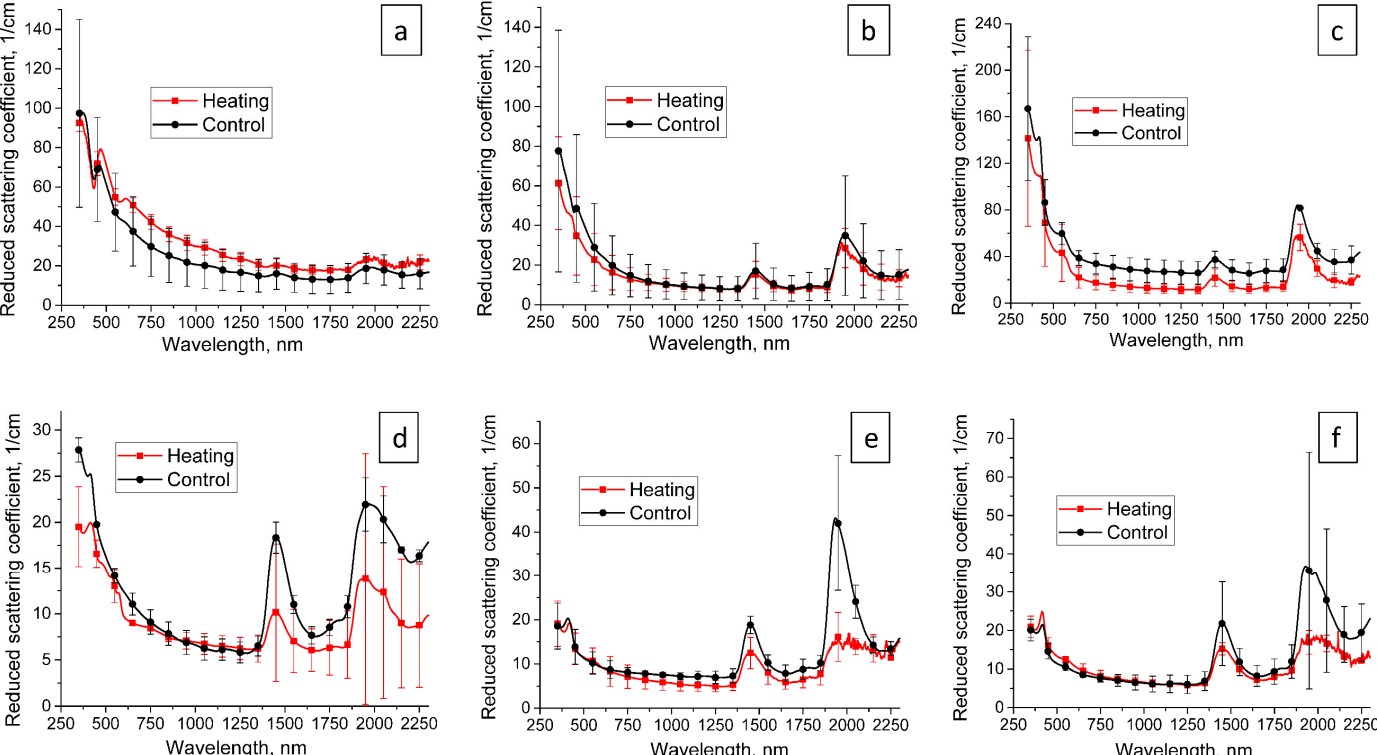

**Figure 8.** Spectra of the reduced scattering coefficient of tumor and surrounding tissues from experimental and control groups: (**a**) skin, (**b**) sub-skin layer, (**c**) tumor capsule, (**d**) tumor top, (**e**) tumor center, and (**f**) tumor bottom.

## 4. Discussion

In experimental studies, the laser irradiance parameters for PPT vary greatly. Specifically, the power density, spot size, and irradiation duration vary from 1.5 to 50 W/cm$^2$, 1 to 5 mm, and 1 to 15 min, respectively (see, e.g., [30]). Thus, the laser intensity used in our study falls within the range used for typical PPT protocols. Our choice is determined by the thermal response of the tissue presented earlier in [12]. This irradiation mode induces heating less than 40 °C without the injection of nanoparticles and does not damage tissues. A decrease in laser intensity decreases the effectiveness of PPT.

The modeling of temperature fields in tumors with embedded gold nanoparticles during laser photothermal therapy and model experiments with intratumorally injected nanoparticles have shown that the heating of nanoparticles and surrounding tissues is a fast process. For example, at the laser pulse duration of 2 ns and the intensity of 5 mW/cm$^2$ at the plasmon resonant wavelength, the temperature increment in the gold nanoparticle reached about 100 °C [31]. In [32], nanoparticles were embedded subcutaneously at a depth of 1 mm, and the maximum temperature at the spot center after laser continuous irradiation with the intensity of 10 W/cm$^2$ during 30 s achieved 65 °C. Numerical simulations of human skin tumor hyperthermia show that effective tumor hyperthermia (42 °C and more) can be performed with the targeted delivery of silicon nanowires synthesized in ethanol at nanoparticle mass concentrations of 3 mg/mL and higher [33]. In [34], numerical simulations show that the temperature required for hyperthermia of tumor with embedded silicon nanoparticles is achieved at concentrations from 0 to 5 mg/mL and intensities from 220 to 170 mW/cm$^2$ for a laser radiation wavelength of 633 nm and from 255 to 225 mW/cm$^2$ for a wavelength of 800 nm.

In this study, we measured the integrated thermal reaction of a living system to the heating of nanoparticles localized in blood vessels and the surrounding space on the skin surface. At the same time, blood perfusion played an important role in this process. It was obtained that the best approximation of the experimental temporal dependence of

the heating is two-exponential temperature growth with different rates (Figure 4). This could be caused by both the geometry of temperature registration and the blood perfusion in the skin. For perfusion rates typical of the skin, perfusion begins to affect the thermal regime only after 60 s of heating [35] and slows down the temperature increase after longer irradiation [36]. Consequently, the fast heating process during $\tau_1 = 0.7 \pm 0.1$ min may be associated with high heating of GNR in the area of their localization in the vessels and interstitial space around the vessels in the tumor capsule and the upper part. During this time, perfusion does not significantly affect tissue heating. Thus, a decrease in the rate of temperature increase at the second stage of heating may be due to the perfusion effect.

As follows from the literature, with an increase in the tissue temperature, the amplitude and position of the absorption peaks of the main chromophores (hemoglobin and water) in the visible and near-infrared (NIR) regions of the spectrum change. For example, it was shown that under slow heating (below 70 °C), part of $HbO_2$ was converted into Hb [18,37]. During heating at 50–80 °C, hemoglobin absorbance changes were partly attributed to oxidative reactions during the formation of met-hemoglobin (metHb) and protein denaturation [38]. A decrease in amplitude and a blue shift of water absorption bands in the skin in vivo after heating from 25 to 60 °C are presented in [18].

As is presented in Figure 9 in the visible spectral range, absorption peaks of oxyhemoglobin ($HbO_2$) are located at about 417 nm (Soret band), 546 nm, and 578 nm (Q bands), and the absorption valley is positioned at about 563 nm. When HbO2 converses to deoxyhemoglobin (Hb), the Soret band shifts to about 423 nm, and a single absorption peak of Hb can be seen at 553 nm.

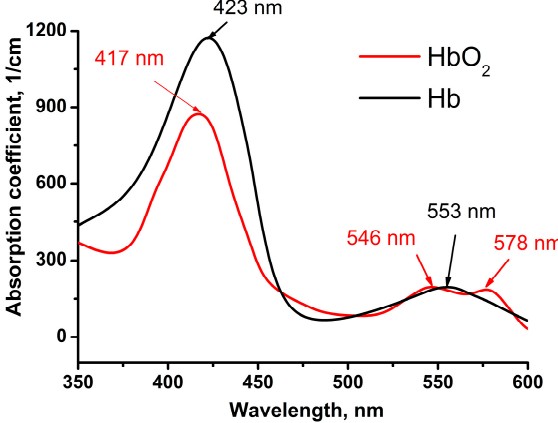

**Figure 9.** Absorption spectra of oxyhemoglobin ($HbO_2$) and deoxyhemoglobin (Hb). Spectra were calculated from the data presented in [39].

It is known that the oxygen affinity of hemoglobin is controlled by the oxygen partial pressure. The diffusion of oxygen from erythrocyte hemoglobin into tissues is due to low oxygen partial pressure in surrounding tissues. In metabolically active tissues, such as tumors, the affinity of hemoglobin to oxygen is lower than in normal tissues. Therefore, in developing tumors, hemoglobin is mainly in a deoxygenated form. An increase in blood temperature changes the ratio of hemoglobin fractions in erythrocytes and causes a shift in the oxyhemoglobin dissociation curve; as a result, more oxygen will be released from hemoglobin [40]. The diffusion reflection spectra measured in vivo on the skin surface in the region of heating (see Figure 6a) clearly show the transition of hemoglobin from an oxygenated to a deoxygenated form.

In our study during PPT, the heat was distributed unevenly inside the tumor; the highest temperature was reached in the localization of the GNRs near the microvessels. However, the optical parameters were measured integrally from the surface of a sample about 5 mm in diameter. Thus, small spectral shifts of the peaks may be indistinguishable. In addition, the spectra were averaged over four samples with different tissue characteristics, concentrations, and localizations of GNRs. However, the change in the peak

amplitudes of the hemoglobin and water absorption bands may indicate a general trend in the spectral changes caused by heating.

Weak absorption by liquid water in the NIR region is due to excitation of higher overtone and combination transitions of water. It was shown that the temperature dependence of the absorption spectrum of liquid water is strongest in the regions of the overtones and combination bands of the water vibrations. As the temperature rises, the intensity of each band shifts from the lower side of the absorption envelope to the higher energy side [41]. It manifests itself in a blue shift of water absorption bands. Temperature-induced blue shift of water absorption peaks in the skin in vivo is observed in Figure 6b.

It can be observed in Figure 7 that depending on the heating temperature of the tissue in vivo, the amplitude of the water absorption bands can either increase or decrease. For example, in skin subjected to heating, edema develops due to the transition of "bound" water to a "free" state; therefore, an increase in the amplitude of water peaks is observed (Figure 7a). In tumor tissue, the general dehydration of the tissue occurs (Figure 7e,f).

Thus, both increasing in hemoglobin absorption bands in the skin and sub skin tissue after heating and decreasing in the amplitude of water absorption bands and blue shift are in good agreement with the results of temperature-induced changes of the skin absorption coefficient presented in the literature.

A comparison of the scattering properties obtained in this study with the results presented in the literature for ex vivo and in vivo tissues shows that our data are qualitatively consistent with the presented trends. A much steeper slope of the diffuse reflectance spectrum of the treated tissue in comparison with intact tissue (Figure 5) can be caused by the destruction of the large scatterers, such as erythrocytes. The results are well agreed with data presented in [18] when mice ear was heated up to 60 °C in vivo. The gentler slope observed in reduced scattering spectra for skin and sub skin layer after the heating, presented in Figure 6a,b, can be caused by a decrease in the concentration of scatterers in skin due to edema development. A steeper slope is observed only for two areas: tumor center and bottom. This effect is accompanied by a significant decrease in water absorption bands that can correspond to an increase in scattering due to dehydration. Coagulation of proteins over a relatively large area occurs only in the skin, which manifests itself in an increase in reflectance signal (Figure 6b) and skin scattering (Figure 8a). Reflectance confocal microscopy confirmed tissue shrinkage during the heating of tissue in vivo from 25 to 60 °C [18].

Temperature-induced changes in different tissues, blood, and water were investigated earlier, in particular in [18–20,37,38,40,41]. However, determining the optical parameters of tissue by measuring the spectra of an entire biological object does not make it possible to assess the contribution of various regions to the measured signal. On the contrary, we evaluated the absorbing and scattering properties of each layer separately. This knowledge can be helpful for accurate dosimetry of photothermal therapy.

Recently, successful clinical trials of gold nanoparticles for plasmonic photothermal therapy of prostate cancer have been published in [42–44]. Clinical safety and efficacy of AuroLase® Therapy (Nanospectra Biosciences, Inc., Houston, TX, USA) including the infusion of AuroShells® nanoparticles and 810-nm laser ablation in the prostate were confirmed in [43,44].

## 5. Conclusions

The changes in the optical properties of different layers of model cholangiocarcinoma in rats doped with GNRs and subjected to PPT relative to control samples were studied in vivo and ex vivo in the spectral range 350–2250 nm. The change of tissue temperature on the surface was from $33.8 \pm 0.3$ up to $54.8 \pm 1.6$ °C. An increase in absorption of hemoglobin and water was induced by edema and hemorrhage in the surface layers. A decrease in the absorption of water inside the tumor indicated tissue dehydration during PPT. Knowledge of temperature-induced changes of tumor optical properties at the different depths will

help to clarify the localization of GNRs to provide correct dosimetry and, thus, increase the effectiveness of PPT.

**Author Contributions:** A.B.B., G.S.T. and E.A.G. conceived and designed the experiments; V.D.G., A.B.B., G.S.T., N.A.N. and E.A.G. performed the experiments; N.G.K. provided nanoparticles; V.D.G. calculated and prepared the original draft of the paper; E.A.G., A.B.B., N.G.K. and V.V.T. reviewed and edited the paper. All authors have read and agreed to the published version of the manuscript.

**Funding:** E.A.G. was supported by the RFBR grant, no. 20-52-56005. V.D.G. was supported by the RFBR grant, no. 19-32-90224, and by the Foundation for Assistance to Small Innovative Enterprises in Science and Technology, grant UMNIK-19/HealthNet-NTI—2019 no. 15929GU/2020 of 07.23.2020 (code 0059878, application (U-65096)). N.G.K. was supported by the Ministry of Science and Higher Education of the Russian Federation, research theme 121032300310-8. The work of A.B.B. and N.A.N. was partially supported by the Saratov State Medical University according to research project no. SSMU-2022-002.

**Institutional Review Board Statement:** The animal study protocol was approved by the Institutional Ethics Committee of Saratov State Medical University (protocol code 6 and date of approval 6 February 2018).

**Informed Consent Statement:** Not applicable.

**Data Availability Statement:** Not applicable.

**Conflicts of Interest:** The authors declare no conflict of interest. The funders had no role in the design of the study; in the collection, analyses, or interpretation of data; in the writing of the manuscript, or in the decision to publish the results.

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
