# Peer review of "Changes in Optical Properties of Model Cholangiocarcinoma after Plasmon-Resonant Photothermal Treatment"

_photonics, doi:10.3390/photonics9030199_

Round 1

Reviewer 1 Report

This study reports the optical properties of tis-21 sue layers of transplanted cholangiocarcinoma and covering tissues in rats without treatment (con-22 trol group) and after PTT using gold nanorods (experimental group). The article is very complete and interesting. It involves several scientific disciplines, which makes it especially attractive. It is to be commended that they have performed both in vitro and in vivo studies, which provide added value. In my opinion, the article could be published if the author considers a series of critical questions that allow justifying the conclusions reached.

  • Authors say: “A diode laser LS-2-N-808-10000 (Laser Systems LLC, Russia) with a wavelength of 808 nm and a power density of 2.3 W/cm2 was used for irradiation. The duration of expo-109 sure was 15 minutes” Why did you choose this wavelength? Other wavelengths in the red and infrared part of the spectrum could work and even provide better results. It would be interesting if you explain in the article the justification for the choice of wavelength.

  • Authors say: Diffuse reflectance spectra were measured in vivo before and immediately after PPT 120 using two commercially available spectrometers. Why did you choose to measure reflectance? The transmittance could provide more conclusive values ​​when considering these wavelengths.

  • Authors say “the absorption band of hemoglobin was registered in the tumor capsule and indicated deoxygenation of hemoglobin”. Authors should explain in more detail the process and the conclusions reached (deoxygenation of hemoglobin). It should show absorption spectra of oxygenated and deoxygenated hemoglobin in order to conclude that it is indeed deoxygenated hemoglobin.

  • Author says “A blue shift of water absorption bands was caused 289 by temperature rise”. Authors should explain in detail the blue shift process of water, and how it is correlated with temperature. It would be useful to have graphs at various temperatures to be able to appreciate the spectral shift that happens (in relation to the temperatura) and justify the conclusions reached. The spectral shift of water could be caused by other factors and not only by temperature. They should explain and justify in detail everything related to the spectral displacement of water as it is critical to be able to draw valid conclusions.

Author Response

All comments are in the Word file.

Reviewer 2 Report

The authors presented their new results about plasmonic photothermal therapy (PPT) via gold nanorods which are in transplanted cholangiocarcinoma of an experimental group of rats and irradiated by laser radiation at the resonance wavelength. Comprehensive analysis of optical properties in the range of 350 – 2250 nm for the tumor, cover tissues and a control group of rats are performed. The explicit PPT effect is revealed. The results are well presented, seem authentic and promising. However, the authors should improve their manuscript accordingly my comments:

  1. You triple used an abbreviation “PTT”. Other abbreviations of plasmonic photothermal therapy are “PPT”. It should be corrected.
  2. Lines 48 – 49, “easy adjustment of plasmon resonance by changing the size ratio [9]”. It will be better to change “size ratio” to “aspect size ratio”. Ref. [9] of Arnida et al. is undesirable here. This article is about gold nanorods with uniform in size and shape and an almost constant aspect ratio. The problem of plasmon resonance adjustment is not solved there. You have to change this reference to a more appropriate.
  3. In the text you should add a mention of the simulated extinction spectrum in Figure 1c.
  4. How many rats were in the experimental and control groups? I can’t find this information in the manuscript.
  5. You should add text about prospects of following clinical studies including experiments with humans. In my opinion, the applied laser intensity of 2.3 W/cm2 seems enough large for permitted therapy. Please, present your estimation and discussion. For example, you might use ANSI Z136.1-2014 – American National Standard for Safe Use of Lasers.
  6. You should add a temperature dependence for the control group in Figure 5 and give explanation of obtained results.
  7. Figures 7 and 8 should be improved. Please, present them in better quality with a higher resolution.

Author Response

All comments are in the Word file.

Round 2

Reviewer 1 Report

Thank you very much for the update and the responses. I think the article can be published in present form.

Reviewer 2 Report

Dear authors,

Thank you so much for your corrections! The manuscript has become noticeably better. I have only one comment left.

In the downloaded file the quality of Figures 7 and 8 stay bad. The same is for Figure 1. Also, the mentioned in the caption red line to Figure 1c doesn’t look color.